# Essential Oils against *Sarcoptes scabiei*

**DOI:** 10.3390/molecules27249067

**Published:** 2022-12-19

**Authors:** Simona Nardoni, Francesca Mancianti

**Affiliations:** 1Department of Veterinary Sciences, University of Pisa, Viale delle Piagge 2, 56124 Pisa, Italy; 2Centro Interdipartimentale di Ricerca “Nutraceutica e Alimentazione per la Salute”, University of Pisa, Via del Borghetto 80, 56124 Pisa, Italy

**Keywords:** *Sarcoptes scabiei*, sarcoptic mange, scabies, essential oils

## Abstract

Herbal remedia are widely employed in folk medicine, and have been more and more often studied and considered in the treatment of several infections. Sarcoptic mange (scabies, when referring to human patients) is a highly contagious skin disease caused by *Sarcoptes scabiei* (sarcoptiformes, Sarcoptinae), an astigmatid mite which burrows into the epidermis, actively penetrating the *stratum corneum.* This parasitosis negatively affects livestock productions and represents a constraint on animal and human health. The treatment relies on permethrine and ivermectine but, since these molecules do not have ovicidal action, more than a single dose should be administered. Toxicity, the possible onset of parasite resistance, the presence of residues in meat and other animal products and environmental contamination are the major constraints. These shortcomings could be reduced by the use of plant extracts that have been in vitro or in vivo checked against these mites, sometimes with promising results. The aim of the present study was to review the literature dealing with the treatment of both scabies and sarcoptic mange by plant-derived agents, notably essential oils.

## 1. Introduction

Sarcoptic mange (scabies, when referring to human patients) is a highly contagious skin disease caused by *Sarcoptes scabiei* (sarcoptiformes, Sarcoptinae), an astigmatid mite which burrows into the epidermis, actively penetrating the *stratum corneum* [1].

Adult mites mate and females lay eggs into the skin. Hatched larvae make short burrows called molting pouches where they molt into nymphs, which become adults.

The parasites occur worldwide, affect more than 150 host species, and show an unexpected epidemiological plasticity, being able to transmit among different hosts [2]. The disease can also develop as a mild infection in animals. It is characterized by itching papules, erythema, scales and alopecia; chronic forms occur with hyperkeratosis and/or exudative crust formation [3,4].

Sarcoptic mange is a highly contagious skin disease, whose transmission occurs via skin-to skin route, or by fomites and/or by contact with an infected environment when frequented by heavily affected hosts [5,6]. Nymphs and females can survive off host up to 21 days, being more resistant than larvae and males [7,8], so biocides and repellents should also be employed in the environment.

Wildlife is considered as more susceptible to sarcoptic mange and outbreaks of infection can lead to high morbidity as well as fatal outcomes [9], especially when naive populations are involved [10]. Sarcoptic mange is an emerging disease [11] and is considered responsible for the decline of wildlife populations [12,13], reducing reproduction and provoking mass mortality events [14,15].

This parasitosis negatively impacts on production animal industries (Davies, 1995; Wei et al., 2019) [16,17], representing a constraint for animal and human health.

Furthermore, scabies is reported by OMS to infect 100–200 million people and has been included among the most neglected tropical diseases [18].

These features largely demonstrate the heavy impact of *S. scabiei* infection on animal husbandry, as well as on human health, when referring both to zoonotic and anthroponotic parasite varieties.

The treatment of these infections in human patients has been recently reviewed [19,20]. Permethrin is considered as the most suitable topical drug, although an in vitro mites’ resistance was reported [21]. Ivermectine in oral formulation is effective, but it is not recommended during pregnancy and in patients with less than 15 kg body weight [22]; moreover, emerging resistance has been observed [23,24]. Furthermore, these molecules do not have an ovicidal effect and more than a single dose should be administered [19]. So toxicity and parasite resistance are major concerns in scabies treatment [25].

Chemical acaricide drugs such as organophosphates, macrocyclic lactones, formamidines, pyrethroids and isooxazolines are registered for veterinary use in the treatment of sarcoptic mange for different animal species [19]. However, an extensive use of these drugs in livestock can represent a threat of the presence of residues in meat and other animal products, as well as environmental contamination.

For these reasons, herbal remedia, widely employed in folk medicine, have been more and more often studied and considered in the treatment of these ailments, both in human patients and in animals. These compounds are, in fact, considered as green products, exerting a low impact on the environment [26,27].

The aim of the present study was to review the literature dealing with the treatment of both scabies and sarcoptic mange by plant-derived agents, notably essential oils (EOs).

## 2. Essential Oils

Essential oils are secondary metabolic end products of plants, stored in different parts, which serve as defense against different pathogens. Almost 17,500 aromatic plant species could be important for pesticidal approaches [28]. EOs mostly consist of a mixture of different terpenes, sesquiterpenes and aromatic compounds such as phenols and phenylpropanes, are responsible for the characteristic fragrance of plants and can be extracted from by distillation, solvents and mechanical squeezing [29]. These compounds exert different biological activities; among these worthy of mention are antibacterial, antiviral, antifungal and antiparasitic effects, referring to a direct action versus pathogens, along with beneficial activities on patients (i.e., antioxidant, anti-inflammatory, immunomodulant effects), which indirectly contribute to healing [30,31]. Furthermore, some molecules among these can show a repellent effect on some acarina [32,33] and insecta [34,35].

Several EOs have been checked in vitro and employed in vivo against *S. scabiei*, from different animal sources and in different animal species. The most relevant results are summarized in Table 1 (in vitro studies) and in Table 2 (in vivo studies).

### 2.1. Melaleuca alternifolia

Tea tree oil (TTO) is extracted from *Melaleuca alternifolia*, a native plant from Australia. Its chemical composition is well defined; the EO contains about 70% terpinen-4-ol, γ-terpinene and α-terpinene and about 15% p-cymene, terpinolene and α-terpineol. TTO exhibits several biological activities as antibacterial, anti-inflammatory properties [46,47], together with antifungal [48,49] and wound-healing effects [50]. The antiparasitic properties of TTO against medically important ectoparasites has been recently carefully reviewed [51].

The mechanism responsible for the anti-mite effect of TTO is not fully elucidated, but the anticholinesterase activity of T4O, 1,8-cineole, a-terpinene, g-terpinene and p-cymene induce a lethal effect by muscular contraction and spastic paralysis of the parasites [52,53]. Furthermore, 1.8 cineole alone exerts a repellent effect on *S. scabiei* var. *cuniculi* and causes alteration in enzyme activity involved in the nervous system of the mites [54].

### 2.2. Azadirachta indica

Neem oil is extracted from *Azadirachta indica,* an autochthonous plant from India, largely employed in folk medicine for its huge spectrum of therapeutical uses [55,56,57]. The most commonly employed EO derives from kernels, although it can be extracted also from fruits [55]. More than 300 compounds have been recognized in the kernel, but the whole composition is far from being determined [56].

Azadirachtin is the main constituent of neem and is found in seed and leaves [58], with a content ranging between 300 and 1.300 mg/kg. This compound is considered responsible for the insecticide activity of neem oil [59]. It shows a repellent effect, is able to induce growth deregulation, reduction in ecdysone levels, alterations in development and reproduction, and damages in the molting process [58]. However, it is reported to have no effects on *Sarcoptes* mites [21], although neem at concentrations ranging from 20% to 60% showed that azadirachtin acts by thinning and weakening the exoskeleton of semi-engorged female of *Rhipicephalus sanguineus,* promoting the absorption of toxics and the rupture of the integument [60].

The acaricide effect against *Sarcoptes* has been ascribed to four fractions, active on larvae of *S. scabiei* var *cuniculi* [61,62], and octadecanoic acid-tetrahydrofuran-3,4-diyl ester [63], able to damage the body wall of mites, interfering with mitochondrial activity and with the oxidative phosphorylation pathway, leading to the parasites’ death [55,64]. Recent transcriptomic studies have shown that some derivatives obtained by structural modification of octadecanoid acid-3,4-tetrahydrofuran diester to yield benzoic acid-2-benzyloxy-3,4-tetrahydrofuran diester strongly enhance the acaricidal activity of the compound, also interfering with the citric acid cycle and fatty acid metabolism [65].

Aqueous and methanolic extracts of neem kernel in ointment in a vaseline vehicle were proven to exert a strong acaricide activity, when administered at dilution 20% in naturally infected sheep over 20 days [66]. Aqueous extracts of neem fruits proved very active, after administration at 25% in pigs affected by sarcoptic mange [67], with a positive outcome in 6 weeks.

Aqueous leaf extracts of neem at 25%, administered every three days for three weeks in experimentally infected rabbits, was highly efficacious in 42 days, without any adverse effect and showing cumulative body weight gain significantly higher in respect to control group [68].

The EO was also assayed as a microemulsion, which proved active at a 10% dilution to kill larval stages of *S. scabiei* [69].

Toxic effects of neem EO consist in contact dermatitis [70], while leaf aqueous extract is sometimes reported to be responsible for anemia, reduced fertility and abortions. Purified components are found to be less toxic for mammals at therapeutic concentrations [71], supporting the potential of this plant as a source for acaricidal [72].

### 2.3. Sygyzum aromaticum

Clove EO is extracted from *Sygyzium aromaticum.* It proves highly active in vitro in killing permethrin-resistant *S. scabiei* var *canis* collected from rabbits and permethrin sensitive mites from pigs at 1.56% within 25 min [73]. Similarly, *S. scabiei* collected from experimentally infected pigs, were killed within 20 min when bio-assayed by contact at 1%; thus, this EO was used as positive control in another recent study [38].

This oil possesses several biological activities as well as antibacterial, antiparasitic and antifungal effects [74], together with immunomodulating properties [31,75].

The effect can be mostly attributed to eugenol, the major component of clove EO, and related compounds. This terpene interferes with cell membranes and organelles both in epidermal and gut epithelia of *Sarcoptes* [73]. Moreover, eugenol was able to kill *Psoroptes cuniculi* mites by regulating the mRNA expression of glutathione S-transferase, catechinic acid and thioredoxin [76] and inhibiting complex I activity of the mitochondrial respiratory chain in the oxidative phosphorylation pathway, with inhibition rate of 60.26% for 100 µg/mL [77]. This compound inhibited 50% (EC_50_) egg hatching at 0.65% and killed mites at 1% in 5 min; the fumigant form also gave interesting results [78]. Eugenol showed an ovicidal effect on *S. scabiei* with EC_50_ at 0.9% [37], being demonstrated as capable of penetrating the surface of eggs and possibly overcoming the need for further treatments [19]. Furthermore, local administration of eugenol should not exceed 0.5% to avoid the risk of skin adverse reactions [79], so the use of EC _50_ should be carefully evaluated.

### 2.4. Cymbopogon Species

Palmarosa EO is extracted from *Cymbopogon martini.* This compound at %, administered by contact, killed *S. scabiei* within 50 min [36]. Geraniol is the main component and seems to have a strong acaricide effect when given by contact, instead of in fumigant form. This terpene was active on both mites and eggs of *P. cuniculi* [78] and also showed an ovicide activity on *S. scabiei* [80] with an EC_50_ of 2%. This concentration is well tolerated for dermal use, being 5.3% of the maximum amount recommended [79], and this EO appears as a promising tool for a safe treatment.

Lemongrass (*Cymbopogon citratus)* EO showed both miticide and ovicide activity, killing the stages at dilution 5% [80]. This EO is rich in nerale and the authors refer to the presence of citral, which has been reported to exert an ovicide action at 4.8% against *S. scabiei* eggs [37]. Furthermore, lemongrass contains geraniol.

Ahibero 1% showed anti-mite activity, killing all stages within 62.53 min. The same concentration was active also in fumigant form. The major compound was limonene [38].

### 2.5. Cinnamomum Species

Cinnamon (*Cinnamomum zeylanicum*) EO has been demonstrated as able to combat *P. cuniculi* [81] and recently proved active against *S. scabiei.* This EO killed 100% of mites at 1%, showing a strong action, both in contact and in fumigant form. Unfortunately, the EO was not ovicide [38], despite the main components being eugenol along with benzyl-benzoate, considered responsible for the action on mites [38].

*Cinnamomum camphora* EO, from different geographical sources (ravintsara from Madagascar and camphorwood from China) contained high amounts of 1.8 cineole and limonene, respectively, but had a miticide activity at 5%, by contact only (ravintsara) and both by contact and in fumigation form (camphorwood) [38].

### 2.6. Ocimum sanctum

Tulsi (*Ocimum sanctum*) EO proved highly effective against *S, scabiei* with an acaricide effect at 0.25%. It is considered as a promising alternative to conventional drugs, although, as for cinnamon, an ovicide effect was never observed. The main component is represented by eugenol (as for clove and cinnamon), along with beta caryophyllene [38].

### 2.7. Litsea cubeba

Litsea (*Litsea cubeba*) EO, previously successfully tested against *Varroa destructor* [82], was effective as ahibero in the same study. The main components are geranial, neral and limonene [38].

### 2.8. Backhousia citriodora

Lemon myrtle (*Backhousia citriodora*) EO showed a similar effect as both Litsea and ahibero, sharing a similar composition [38]; moreover, this plant is considered a main source of citral [83]. A *B. citriodora* EO, containing high amounts of this compound, showed a strong repellent effect against ticks [84]. The components of lemon myrtle EO in different studies varied widely, thus corroborating the huge variability in EO composition of different plants within the same botanical species and, subsequently, in their biological effects.

### 2.9. Citrus limon

Lemon (*Citrus limon*) EO is an interesting remedium for rabbit sarcoptic mange. A 10% solution killed in vitro 100% of mites and, when topically administered at 20% on lesions once a week for four weeks, led to complete clinical and etiological recovery in the affected subjects, with better results in respect to deltamethrin-treated controls. Furthermore, the oxidative stress profile of lemon-treated mites appeared strongly affected. Finally, the treated animals had better productive performance in respect to control groups [41]. The main components of lemon EO is limonene, active in disrupting the respiratory system function of acari [85].

### 2.10. Eucalyptus Species

Eucalyptus (*Eucalyptus radiata*) EO showed a LD_50_ in 20 min at 10% on *S. scabiei.* The main component was 1.8 cineole [36] and it was more effective by fumigation rather than by contact.

After 3 h of contact, 200 mg/mL of *Eucalyptus globulus* showed good in vitro acaricidal efficacy compared to untreated controls; promising results were given also by *Dodonea angustifolia, Millettia ferruginea* and *Euphorbia abyssinica* EOs [86].

Eucalyptus EO is reported to exert an anti-inflammatory effect, along with immunomodulant properties [31].

### 2.11. Pelargonium asperum

Geranium (*Pelargonium asperum*) EO showed a LD_50_ in 20 min at 5% on *S. scabiei*, when administered by contact. The main components were citronellol and geraniol [36].

### 2.12. Lavandula angustifolia

Lavender (*Lavandula angustifolia*) EO showed a LD_50_ in 20 min at 10% on *S. scabiei.* The main components are represented by linalool and linalyl-acetate [36]. Linalool did not interfere with the development of embryos [37], but proved active against *P. cuniculi,* while linalyl-acetate did not show any effect [87].

Lavender is also reported to exert an immunomodulatory effect [31].

### 2.13. Citrus aurantium amara

Bitter orange (*Citrus aurantium amara*) EO yielded similar results to those given by lavender, showing a similar composition [36].

### 2.14. Lippia multiflora

Essential oil from *Lippia multiflora* was checked in vivo and proved highly effective in patients affected by scabies, when administered at 20% for five days, giving 100% recovery [42]. The main components are linalool, geraniol and limonene. These compounds have been reported as active against *S. scabiei*, as reported above [37,85].

### 2.15. Cedrus deodara

Himalayan cedrus (*Cedrus deodara*) EO was topically administered to lambs naturally infected by *S. scabiei* for 10 days and was proven to enhance the activity of benzyl-benzoate, leading to complete recovery [43].

### 2.16. Pongamia pinnata

Sulphur-karanj (*Pongamia pinnata*) EO 10% was employed in a successful treatment of sarcoptic mange in goats, six times every 3 days [44].

### 2.17. Jatropha curcas

*Jatropha curcas* has been reported as effective against scabies [44]. This EO is commonly employed in India and, in a further trial, administered together with ascorbic acid, mainly *J. curcas,* but also *C. deodara* and *P. glabra*, also showed a therapeutic effect on ovine sarcoptic mange and significantly improved wool production and meat quality [44].

### 2.18. Elsholtzia densa

*Elsholtzia densa* EO was in vitro checked against *S. scabiei*: 0.981 mg/mL killed mites after 24 hrs. contact. The main components are represented by 4-pyridinol and thymol [88].

The miticide activity of EOS, as previously reported, has been mainly ascribed to some terpenes. Carvacrol is contained in EOs from Lamiaceae [89] and shows several biological effects. It was strongly active against *P. ovis*, both in vitro and in vivo [90], as well as against *S. scabiei* eggs [37], with an EC_50_ of 0.5%. This concentration can be well tolerated for skin use [79]. Thymol was also recently used in vitro against *S. scabiei,* resulting in high toxicity, interfering with the energy metabolism and nerve conduction of the mites [88].

*Cedrelopsis grevei, Curcuma longa, Pinus pinaster* and *Lantana camara* EOs at 10% at did not yield any miticide activity [38], nor did Juniperus oxycedrus EO [36].

## 3. Plant Extracts

Several plant extracts have been checked for the treatment of scabies and sarcoptic mange, apart from aqueous neem (see above). They are mostly used in folk medicine and have been recently revised by Akram et al. and by Shiven et al. [20,91], highlighting the interest of this topic to the scientific community.

However, worth mentioning is an in silico and in vitro study on the efficacy of ethanol extracts of *Acacia nilotica* and *Psidium guajava* versus *S. scabiei* var. *cuniculi.* The rationale of the study was the evaluation of these compounds against the aspartic protease of the mites, identified as therapeutic target. *A. nilotica* showed a miticide efficacy comparable to permethrin, and better than the effect of *P. guajava* [91].

## 4. Conclusions

EOs can be considered from a green perspective, and represent an easy tool, being biodegradable, with low ecotoxicity and minimal environmental residual activity due to their high volatility [92]. Their application in the treatment of scabies seems versatile, having shown that some EOs (i.e., cinnamon and tulsi) have very low MIC values, others (lemongrass), along with some terpenes (geraniol, eugenol and citral), also exhibit ovicidal effects, while some are effective both by contact and in fumigant form (ahibero) [38].

The future trends for the acaricidal use of EOs and/or for their most active components involve the development of mixtures. The synergistic effect of these molecules would allow decrease in the amount of administered compounds, increasing the sphere of action. In fact, EOs being complex chemical compounds, they can have a potential toxicity which should be taken into account, mostly in particular physiological conditions such as pregnancy [93].

Furthermore, although safe and easily administrable, the main drawbacks rely on the great differences in composition of plant derived remedies within the same botanical species, making it necessary to provide a standardization of EOs, now available for only a few.

The study of EOs in the management of sarcoptic mange and scabies would open new perspectives in the treatment of these skin infections, mostly when ovicide compounds are evaluated, significantly reducing the number of drug administrations. Furthermore, this green perspective would allow reduction in the toxicity of acaricide treatments, the emergence of drug resistance in arthropods and, when livestock are involved, the presence of residues in animal products.

## Figures and Tables

**Table 1 molecules-27-09067-t001:** Essential oils (EOs) active up to 5%.

EO	Mite	References
Tea tree oil 5%	*S. scabiei* var *suis*	[36]
Clove 1%	*S. scabiei* var *suis*	[36]
Palmarosa 1%	*S. scabiei* var *suis*	[36]
Geranium 5%	*S. scabiei* var *suis*	[36]
Lemongrass 4.8%	*S. scabiei* var *cuniculi* mites and eggs	[37]
Ahibero 1%	*S. scabiei* var *suis*	[38]
Cinnamon 0.25%	*S. scabiei* var *suis*	[38]
Ravintsara 5%	*S. scabiei* var *suis*	[38]
Camphorwood 5%	*S. scabiei* var *suis*	[38]
Tulsi 0.25%	*S. scabiei* var *suis*	[38]
Litsea 1%	*S. scabiei* var *suis*	[38]
Lemon Myrtle 1%	*S. scabiei* var *suis*	[38]

**Table 2 molecules-27-09067-t002:** In vivo successful administration of essential oils (EOs) and plant extracts (PE).

EO	Patient	Posology	References
Tea tree oil 5% + benzyl benzoate	human patient	11 administrations	[39]
Tea tree 5% in cream	children	once a week for 2–3 weeks	[40]
Lemon oil 20%	rabbit	once a week for 4 weeks	[41]
*Lippia multiflora 20%*	human patients	five days	[42]
*Cedrus deodora*	lambs	ten days	[43]
*Cedrus deodora +* ascorbic acid	sheep	once daily for 5 days	[44]
PE			
*Pongamia pinnata* 10% + sulfur	goat	six times every three days	[45]
*Jatropha curcas +* ascorbic acid	sheep	once daily for 2 days	[44]
*Pongamia glabra* + ascorbic acid	sheep	once daily for 3 days	[44]

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
