# Peer review of "Essential Oils against Sarcoptes scabiei"

_molecules, 2022, doi:10.3390/molecules27249067_

Round 1

Reviewer 1 Report

The aim of the present study was to review the literature dealing with the treatment of both scabies and sarcoptic mange by plant-derived agents, notably essential oils. It is a relatively well-prepared manuscript that is suitable for publication, but only after the following modifications:

1. Table 2. In vivo administration of essential oils (EOs). Pongamia pinnata and Jatropa seed oils cannot be considered EOs. Please correct the description of the tapula.

2. It would be useful not only to state the effectiveness, but also to select the most effective EOs (extracts) using a suitable meta-analysis and on the basis of suitable criteria. Discuss the effectiveness of substances, extracts or EOs also in terms of possible implementation (based on availability, price, health safety).

3. In conclusion, it would be useful to outline other possible research directions.

Author Response

The aim of the present study was to review the literature dealing with the treatment of both scabies and sarcoptic mange by plant-derived agents, notably essential oils. It is a relatively well-prepared manuscript that is suitable for publication, but only after the following modifications.

The Authors would thank the reviewer for his/her observations, and will do their best to answer properly to all outlined concerns

  1. Table 2. In vivo administration of essential oils (EOs). Pongamia pinnata and Jatropa seed oils cannot be considered EOs. Please correct the description of the tapula.

Captions have been modified in the text, accordingly to the reviewer’s suggestions

2. It would be useful not only to state the effectiveness, but also to select the most effective EOs (extracts) using a suitable meta-analysis and on the basis of suitable criteria. Discuss the effectiveness of substances, extracts or EOs also in terms of possible implementation (based on availability, price, health safety).

and

3. In conclusion, it would be useful to outline other possible research directions.

The Authors strongly enhanced the Conclusions section, following the reviewer’s suggestions, while a meta-analysis was beyond the aim of the present study,  which was intended to be a scoping review, rather than a systematic review.

Reviewer 2 Report

General

This review by S. Nardoni and F. Mancanti about the potential use of essential oils (Eos) against Sarcoptes scabiei in Human and other animals shows the current data on this interesting topic. To my knowledge, there is no other recent review on the same subject.

The review is well written, pertinent and clear.

The abstract is a short introduction of the paper with a last sentence exposing the aim of the review. It does not reflect very well the content of the paper.

The conclusion is rather brief. Authors may give their point of view about the more promising Eos for scabies and mange treatment. They also could elaborate about the advantages and disadvantages of Eos. In particular, they could comment on « the green perspective » as Eos are indeed of natural origin but are also complex chemical products not devoid of toxicity.

Specific

Line 53 : According to Ref 18, the estimated global prevalence of scabies is 100-200 million cases.

Line 79 : Maybe, replace « 17,500 aromatic plant species are important » by « 17,500 aromatic plant species could be important ».

Lines 118-122 : The sentence is too long and contradictory. Please rephrase to make it clearer.

Lines 123-128 : It is unnecessary to mention the fractions and the process of extraction of Eos.

Line 153 : Please replace « recenty » by « recent ».

161-162 : Please correct as following : 60.26% for 100 µg/mL.

Line 182 : Replace » Andriantoanirina » by « Andriantsoanirina ».

Line 204 : Delete « Andria » after [38].

Line 204 : The use of the word « however » in this sentence is not clear for me.

Lines 206-208 : The interesting point of the variability in composition of the Eos from the same botanical species according to various factors could be included in the Conclusions section. Otherwise, please rephrase the paragraph 2.8 to make it clearer.

Line 246 : Please correct « benzil-benzoate » by « benzyl-benzoate».

Author Response

General concerns

This review by S. Nardoni and F. Mancianti about the potential use of essential oils (Eos) against Sarcoptes scabiei in Human and other animals shows the current data on this interesting topic. To my knowledge, there is no other recent review on the same subject. The review is well written, pertinent and clear.

The Authors would thank the reviewer for his/her kind comments

The abstract is a short introduction of the paper with a last sentence exposing the aim of the review. It does not reflect very well the content of the paper.

The abstract has been modified, as kindly suggested

The conclusion is rather brief. Authors may give their point of view about the more promising Eos for scabies and mange treatment. They also could elaborate about the advantages and disadvantages of Eos. In particular, they could comment on « the green perspective » as Eos are indeed of natural origin but are also complex chemical products not devoid of toxicity.

Conclusion section has been enhanced and modified, accordingly to reviewer's observations

Specific comments were all addressed directly in the body of the text

Reviewer 3 Report

This is a review of essential oils used for Sarcoptes infection.

In introduction, there are too many paragraphs on the biology of mites, not very related to the theme of this review.

Table 1 and Table 2 are not enough informative. There are much more essential oils had been tested, but the authors just pick some oils in the tables, should explain why. And the tables should specify the efficacy of each oil,  the outcome of patient after the treatment of each essential oil.

The article emphasized the ovicidal action of essential oils  in conclusion, but many published ovicidal activities of essential oils/compounds haven’t been discussed in the article.

My main reservation about the article is that is it largely just a list of various essential oils that have been published. A review article needs some critical commentary. Which essential oil or oils are more promising for use in the future?  Despite the in vitro and in vivo studies, are any of these oils have already used in the medical community? Are there any limitation for the usage of essential oils?   What are the next steps? 

Minor comments:

-P5 L14 replace “citrale” by “citral”

-P5 L38 delete “Andria)”

-P6 L11 “in vitro” italic?

Author Response

This is a review of essential oils used for Sarcoptes infection.

The Author would thank the reviewer for her/his valuable observations, very useful to enhance the scientific value of the work.

In introduction, there are too many paragraphs on the biology of mites, not very related to the theme of this review.

The Introduction section has been shortened, accordingly to the reviewer’s suggestion. Some parts about mites biology were kept in the text, in order to make it easily comprehensible for all Molecules readers, including the scientists whose field of interest may not be strictly related to parasitology, also.

Table 1 and Table 2 are not enough informative. There are much more essential oils had been tested, but the authors just pick some oils in the tables, should explain why. And the tables should specify the efficacy of each oil, the outcome of patient after the treatment of each essential oil.

Captions of Table 2 have been modified according to the reviewer’s suggestion, Table 1 was composed including the essential oils showing a biological activity up to 5% concentration

The article emphasized the ovicidal action of essential oils in conclusion, but many published ovicidal activities of essential oils/compounds haven’t been discussed in the article.

Unfortunately, the Authors were not able to find any other about an ovicide activity of other essential oils, specifically directed against Sarcoptes scabiei

My main reservation about the article is that is it largely just a list of various essential oils that have been published. A review article needs some critical commentary. Which essential oil or oils are more promising for use in the future?  Despite the in vitro and in vivo studies, are any of these oils have already used in the medical community? Are there any limitation for the usage of essential oils?   What are the next steps? 

Conclusion section has been enhanced as requested, and future trends, as well as shortcomings for EOs use have been added in the text

Specific minor comments have been amended modifying the body of the text, as suggested

Round 2

Reviewer 1 Report

The authors corrected the manuscript according to the reviewers' comments, I have no further comments.

Author Response

We would thank the reviewer for his/her work, and for all the valuable comments, that helped so much in improving the scientific quality of the manuscript.

The Authors

Reviewer 3 Report

Line 274-278 In conclusion part, should add references to this sentence. 

Author Response

The appropriate reference was added in the text,  as recommended.

We would like to thank so much the reviewer for his/her work, that helped us to improve the scientific value of the work.

The authors